# Solvation Structure and Ion–Solvent Hydrogen Bonding of Hydrated Fluoride, Chloride and Bromide—A Comparative QM/MM MD Simulation Study

**Thomas S. Hofer**

Theoretical Chemistry Division, Institute of General, Inorganic and Theoretical Chemistry, Center for Chemistry and Biomedicine, University of Innsbruck, Innrain 80-82, A-6020 Innsbruck, Austria; t.hofer@uibk.ac.at;
Tel.: +43-512-507-57111; Fax: +43-512-507-57199

**Abstract:** In this study, the correlated resolution-of-identity Møller–Plesset perturbation theory of second order (RIMP2) ab initio level of theory has been combined with the newly parameterised, flexible SPC-mTR2 water model to formulate an advanced QM/MM MD simulation protocol to study the solvation properties of the solutes $F^-$, $Cl^-$ and $Br^-$ in aqueous solution. After the identification of suitable ion–water Lennard–Jones parameters for the QM/MM coupling, a total simulation period of 10 ps (equilibration) plus 25 ps (sampling) could be achieved for each target system at QM/MM conditions. The resulting simulation data enable an in-depth analysis of the respective hydration structure, the first shell ligand exchange characteristics and the impact of solute–solvent hydrogen bonding on the structural properties of first shell water molecules. While a rather unexpected tailing of the first shell ion–oxygen peak renders the identification of a suitable QM boundary region challenging, the presented simulation results provide a valuable primer for more advanced simulation approaches focused on the determination of single-ion thermodynamical properties.

**Keywords:** anionic hydration; hydrogen bonding; aqueous solution; ab initio; molecular dynamics; QM/MM





## 1. Introduction

Liquids and solutions are omnipresent states of matter in nature [1,2] and are oftentimes regarded as one of the most versatile tools available in chemistry [3–5]. Processes occurring in the liquid state can be steered in numerous ways by adjusting parameters of the reaction environment. The latter can be achieved for instance by comparably simple measures such as variations in temperature and pressure, or by more complex strategies such as the addition of cosolvents [6,7]. In particular, the addition of salt to a polar solvent may have a strong impact on the overall properties of the liquid [8–10], mainly due to the long-ranged nature of the comparatively strong Coulomb interactions [11], which is linked to a number of prevalent problems in physical and chemical sciences [12,13].

In particular, the requirement of electroneutrality makes it impossible to independently investigate the preferential solvation of cat- or anionic species via experimental means [13,14]. As a consequence, the structural and dynamical properties of any ionic species cannot be investigated without the impact of their respective counter ions. This limitation is extended to the field of ion thermodynamics [13], i.e., it is not possible to separate measured thermodynamic properties such as the solvation free energy and the associated enthalpic and entropic contributions of a dissolved salt into the respective cationic and anionic portion. While this is a known limitation often associated to calorimetric measurements, it can be shown that this shortcoming is directly linked to the potential of an electrochemical cell [15]. Also in this case, the electroneutrality constraint prevents an experimental determination of absolute single-electrode potentials.

An increasingly successful alternative to experimental approaches are theoretical calculations [16], foremost chemical simulation methods based on the Monte-Carlo (MC) and molecular dynamics (MD) frameworks [17–19]. These approaches aim at the description of a target system by collecting a large number of individual configurations associated to a thermodynamic ensemble [20,21]. In contrast to experimental measurements, the investigation of a single ionic species in the absence of counter ions is perfectly feasible using these simulation approaches [22,23]. In particular, simulation protocols based on a quantum chemical [24–27] description of the ion–solvent interactions have proven as a versatile approach to study the manifold properties of solvated ionic systems [28–31].

In addition to the determination of the respective structure and dynamics of the solvation complex, suitable protocols to also investigate thermodynamic quantities have been formulated [23,32–39]. These methods have been applied with remarkable success to characterise the hydration properties of monovalent cations including a consistent evaluation of the associated single-ion solvation free energy. The latter can then be directly related to the absolute single-electrode potential of the associated half-cell reactions [13]. Based on these results, it possible to anchor the entire series of electrochemical potentials without the need of a predefined reference such as the standard hydrogen electrode.

Despite the consistent results achieved in previous studies focused on cationic hydration [38,40], further validation of these results is required. Specifically, the determination of solvation free energies for anionic species appears as a crucial requirement to supplement the data obtained in the cationic case. However, the interaction between anions and the solvent water via hydrogen bonding to the anionic solute is much more intricate in the theoretical description for a number of reasons. First, hydrogen bonded interactions as observed in ion–solvent binding are more challenging in their description compared to the conceptually simpler charge–dipole interaction prevalent in the cationic case. Moreover, an anionic solute introduces excess electron density into the system which typically is more sensitive to shortcomings in the theoretical description than cations with a similar (but opposite) net charge. While the majority of theoretical calculations rely on methods based on density functional theory (DFT) [26,27], certain shortcomings such as the calibration to empirical data have been shown to lead to inaccurate results even in case of simpler cationic hydrates [41–43] and even pure water [44,45]. While the application of empirical dispersion corrections [46–49] may compensate some of these inaccuracies [50], the combination of a quantum mechanical (QM) calculation method with forcefield-like correction terms has also been criticised as combing the worst of both worlds, i.e., the demanding computational effort of QM approaches with the dependence on empirical parameters inherent to a forcefield (FF) description.

While being computationally much more expensive, methods based on perturbative approaches such as the Møller–Plesset perturbation theory of second order (MP2) [51] proved as an accurate and versatile alternative approach to describe the molecular interactions in QM-based studies. Especially in conjunction with the resolution-of-identity (RI) framework [52–54] capable of greatly accelerating the calculations with only a very minor reduction in accuracy, RIMP2-based hybrid quantum mechanical/molecular mechanical (QM/MM) simulations became feasible on modern computational infrastructure. Pioneered by the Nobel laureates Martin Karplus, Michael Levitt and Arieh Warshel [55] in the regime of biomolecular simulations [56–60], hybrid QM/MM MD simulations [56,58,61–63] have been applied with large success in the description of solvated ionic systems [28,29,64,65]. In this approach, the chemically most relevant part, comprised of the solute and solvent molecules in its immediate surroundings, is treated via a suitable quantum mechanical (QM) approach. In contrast, the interactions in the remainder of the simulation system are treated based on efficient molecular mechanical (MM) interaction models. That way, QM/MM approaches exploit the accuracy of a QM description inherently accounting for many-body contributions such as polarisation and charge-transfer in conjunction with the high efficiency of MM interaction potentials.

A large variety of MM potential models have been developed in the past, which is particularly true in case of the solvent water [66–68]. Based on arguments provided by statistical thermodynamics and quantum mechanics, the majority of these solvent models employ a rigid description of water, i.e., the individual water molecules can only perform translational and rotational motion while the respective intramolecular degrees fo freedom are subject to holonomic constraints [69]. While a number of highly successful water models have been formulated based on this strategy, a rigid-body description of the solvent does not provide an ideal basis to represent the complex hydrogen bonding pattern associated to anionic hydration. This is due to the fact that the latter induces a symmetry break in the water molecules, leading to variations in the O–H bond lengths and oftentimes to an adjustment of the intramolecular H–O–H angle. The latter can only be described if a fully flexible description accounting for explicit hydrogen motion is employed.

Although a number of water models accounting for molecular flexibility are available in the literature, the majority of them are not suitable for application in a QM/MM MD simulation study of an anionic solute. Recently, an improved flexible water potential providing a reliable representation of the intramolecular water geometry along with improved vibrational and dielectric properties has been presented [70]. In the present study, this new solvent model is applied for the first time in a RIMP2-based QM/MM MD simulation to investigate the solvation properties of fluoride, chloride and bromide in aqueous solution.

## 2. Methodology

In this section, the simulation strategy applied in the study of aqueous $F^-$, $Cl^-$ and $Br^-$ is outlined. The setup and simulation protocols are largely inspired by the previous investigations of $Li^+$, $Na^+$ and $K^+$ in aqueous solution [38,40]. In a QM/MM simulation setting, the chemically most relevant region of the system, in this case the ionic solute and the coordinating solvent molecules of the first hydration layer, is treated using the chosen QM method, while classical MM interaction potentials are considered sufficiently accurate to model the remainder of the simulation system. Prior to the execution of the QM/MM MD simulation, suitable interaction potentials for the solute–solute and solute–solvent interaction beyond the QM region have to be defined. While a large number of adequate solvent–solvent potentials are available in the literature, suitable ion–water potentials describing the interaction between the QM solute and water molecules compatible with the QM treatment have to be identified. In this work, new interaction parameters for the ion–water interaction have been derived, which is discussed in the following subsection. Next, the simulation protocol is outlined followed by a description of the applied analysis strategy.

### 2.1. Ion–Water Interaction Potentials

Although in a QM/MM study, the immediate interaction between the ionic solute and solvent molecules included in the QM region is described based on the selected quantum chemical calculation method, it is still required to provide an ion–solvent potential to account for interactions between the ionic solute located in the center of the QM region and the classically treated water molecules assigned to the MM zone. While it is possible to avoid the non-coulombic part of this ion–solvent potential if an enlarged QM region encompassing at least two full layers of hydration is employed [71,72], such a simulation setup is prohibitively expensive when aiming at the application of a correlated ab initio method as done in this work.

A key requirement in the selection of the ion–water potential model is a qualitatively correct description of the hydration complex compatible with that of the applied QM method. This is, to some extent, a conceptual dilemma since the structural description resulting from the QM/MM description is only known after the respective simulation is completed, while a suitable ion–solvent interaction potential has to be provided for the execution of said QM/MM simulation. To avoid this situation, suitable potential data are obtained from a series of classical simulations with varying potential settings, that are compared to data of previous QM/MM [73,74] and Car-Parrinello molecular dynamics

(CPMD) simulations [75–79]. In addition, results obtained from experimental investigations such as X-ray absorption fine structures (XAFS) and neutron diffraction with hydrogen isotope substitution (NDIS) also serve as a suitable reference [76,80–82]. While a number of experimental studies are reported in the literature, the solvation properties of halogen ions seem to be quite sensitive to concentration effects (see for instance Wallen et al. [83]). In addition, many experimental studies of salt solutions employing X-ray diffraction report data measured at comparably large concentrations [84–89], highlighting even the formation of solvent-separated ion pairs [90]. However, in the context of this study, treating a single solute ion solvated by 2000 water molecules, reference data for dilute solutions are preferred.

Since the $Cl^-$ ion is an important solute in the context of biomolecular simulations, a large number of potentials are reported in the literature. In contrast, $F^-$ and $Br^-$ are less relevant in this context and, hence, only a smaller number of potential parameters are available. To treat all considered solutes on the same basis, new interaction parameters have been generated for the three ions by performing a series of purely classical MD simulations of a negatively charged ion ($q = -1.0e$) in aqueous solution. A total of 42 individual MD runs have been executed, each with a different setting of the ion–O Lennard–Jones (LJ) parameters [91,92] as depicted in Figure 1a.

The simulation systems in this preliminary potential evaluation were composed of the solute plus 1000 SPC/E water molecules [93] contained in a cubic, periodic simulation cell with a side length of approx. 31.1 Å corresponding to a target density of 0.997 kg dm$^{-3}$. Long-range coulombic contributions have been accounted for via the reaction-field approach [94] employing a relative static permittivity of 78, with the general cutoff distance set to 12.5 Å. The equations of motion have been integrated via the velocity-Verlet algorithm [95]. To achieve a time step of 1.0 fs, the M-SHAKE/M-RATTLE [69,96] algorithms have been applied. Thermal control was realised via the Berendsen algorithm [97] with the target temperature and relaxation time set to 298.15 K and 0.1 ps. For each tested LJ parameter pair, an equilibration for 30,000 MD steps (30 ps) has been carried out, followed by 50,000 steps of sampling (50 ps) collecting data every 25th MD step.

By comparing the location of the resulting first shell peak maximum in the ion-O radial distribution function and the average coordination number with reference data from the literature [73–83], the best potential parameters for the $F^-$, $Cl^-$ and $Br^-$ could be identified (see Figure 1b and Table 1).

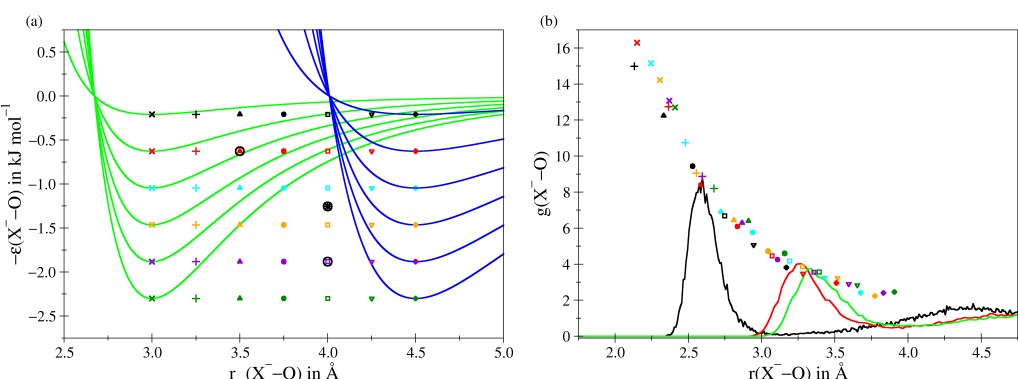

**Figure 1.** (**a**) Overview of the 42 ion–oxygen Lennard–Jones parameter combinations employed to identify suitable interaction potentials for the description of aqueous $F^-$, $Cl^-$ and $Br^-$. (**b**) Maxima of the first shell peak in the ion–oxygen RDF determined for a hydrated ion with charge $-1.0e$ in combination with the 42 Lennard–Jones parameters. The RDFs obtained for $F^-$ (black), $Cl^-$ (red) and $Br^-$ (green) have been selected by comparing the resulting ion–oxygen RDFs and the the coordination number in the first hydration shell to data provided in the literature. The respective parameter sets are highlighted via the black circles and are listed in Table 1. In the case of $Cl^-$ an additional simulation employing an $\varepsilon$-value of 1.2552 kJ mol$^{-1}$ proved necessary.

**Table 1.** Lennard–Jones potential parameters $\varepsilon(X^--O)$ in kJ mol$^{-1}$ and $\sigma(X^--O)$ in Å of the ion–oxygen potential describing the interaction between the QM solute and water molecules in the MM zone. The respective minimum distance $r_m(X^--O)$ in Å is listed as well (see also Figure 1).

| | $\varepsilon(\mathbf{X^--O})$ | $\sigma(\mathbf{X^--O})$ | $r_{\mathbf{m}}(\mathbf{X^--O})$ |
|---|---|---|---|
| F$^-$ | 0.6276 | 3.1181 | 3.5 |
| Cl$^-$ | 1.2552 | 3.5636 | 4.0 |
| Br$^-$ | 1.8828 | 3.5636 | 4.0 |

*2.2. QM/MM MD Simulation Protocol*

The QM/MM MD simulations carried out in this work are following the general setup of previous successful simulation studies of solvated Li$^+$, Na$^+$ and K$^+$ [38,40]. The QM/MM treatment is based on the highly successful quantum mechanical charge field molecular dynamics (QMCF MD) framework [71,72]. In this approach, the partial charges of the MM atoms are included in the Hamiltonian of the QM calculation in form as an external potential [62,98,99]. It was shown that this framework referred to as electrostatic embedding greatly improves the treatment of the QM zone over calculations in an artificial in vacuo environment, thereby enhancing for instance the description of hydrogen bonds passing through the QM/MM interface [100,101]. In addition, polarization and charge transfer effects occurring in the QM treatment are incorporated into the QM/MM coulombic coupling by employing QM-derived partial charges that are updated in every simulation step [71,72]. In order to ensure compatibility with the partial charges of the employed MM model [70], Mulliken populations [102,103] proved as an adequate choice [100,101] despite their general criticism of showing a pronounced basis set dependence. To avoid the latter, it is crucial to employ balanced basis sets, i.e., assigning diffuse and polarization functions in an equal matter to all atoms in the systems including the oftentimes neglected hydrogen atoms.

In order to account for the distortion of the molecular geometries of water molecules in the vicinity of the solute, the flexible SPC-mTR2 water model [70] has been employed. The latter is a re-parametrization of the SPC-mTR (simple point charge modified Toukan–Rahman) water potential [104], in which the charges, the equilibrium bond length and angle, and further parameters have been adjusted to provide a more reliable representation of the intramolecular water geometry while at the same time the vibrational and dielectric properties were improved.

An enlarged simulation cell containing the ion and 2000 water molecules in the isobaric-isothermal ensemble (NPT) under periodic boundary conditions (side length approx. 39.4 Å) was employed in each case. Also in case of the QM/MM MD simulations, the long-range electrostatic contributions were treated via the reaction-field approach [94] utilizing an extended cutoff radius of 18.0 Å. The MD time step in the velocity-Verlet integration [95] was set to 0.5 fs to describe explicit hydrogen motion. The Berendsen thermostat and manostat algorithms [97] have been employed to maintain the target conditions of 298.15 K and 1.013 bar, the associated relaxation times were set to 0.1 and 0.5 ps, respectively.

As in the previous studies of Li$^+$, Na$^+$ and K$^+$, resolution-of-identity Møller–Plesset perturbation theory of second order (RIMP2) [51–54] has been applied to describe the interactions inside the QM region as implemented in the TURBOMOLE package [105]. Since, in contrast to cationic solutes, anions display an excess of electrons, the use of larger basis sets is recommended. In this study, 6-31++G(d,p) bases [106–109] were assigned to the atoms of the water molecules, while the larger TZVP basis set [110] was employed for the description of the anions. In addition, suitable auxiliary basis sets [53] for the resolution-of-identity (RI) treatment were applied.

The size of the QM region is set to a radius to include the solute plus its entire first solvation layer based on the center of mass of the individual water molecules. Since the ideal setting is unknown prior to starting the simulation study, the radius is typically

selected based on the minimum distance separating the first and second solvation shell in the ion–O RDF obtained from purely classical MD simulation. This strategy proved adequate in a number of similar QM/MM studies. The radius determining the size of the QM region was set to 3.2 Å in the case of F$^-$, while for both Cl$^-$ and Br$^-$ a value of 4.1 Å was determined from the classical simulation. The reason for the similar QM radii in case of the latter two systems can be explained by the fact that the respective $\sigma$-values in the ion–O Lennard–Jones potential are similar, while the $\varepsilon$-parameters representing the depth of potential are different. This not only influences the attractive part of the potential, but also increases the steepness of the repulsive branch, thereby leading to a larger ion–O distance in the Br$^-$ case. In order to enable a smooth transition of molecules between the QM and MM zone, a small buffer region of 0.2 Å thickness is considered in which the forces are continuously changed from the QM to the MM contribution and vice versa.

The simulation systems for each solute were equilibrated in the NPT ensemble via classical MD for a total of 100,000 MD steps (50 ps) at target conditions. Considering the much larger size of the MM zone compared to the QM region, all molecules associated to the MM treatment are well-equilibrated after this procedure. Next, a re-equilibration for a minimum of 20,000 steps (10 ps) was carried out using the QM/MM setup, followed by 50,000 MD steps of sampling, resulting in a total simulation time of 25 ps. Since, in the construction of the ion–oxygen Lennard–Jones potentials special care was given to replicate the ideal ion–water distances and the associated coordination numbers, the only required equilibration is associated to the intramolecular degrees of freedom. The impact of polarization and many-body effects then has an influence on the first shell ligand exchange rate. Since monovalent ions are known to quickly exchange the first shell ligands within the picosecond timescale [111], the re-equilibration period of 10 ps can be considered as sufficient.

*2.3. Analysis*

The solute–solvent interactions were analysed in terms of ion-water radial distribution functions (RDFs), thereby also comparing the differences observed between the classical and QM/MM-based simulations. In order to provide further information about the composition of the first solvation layer, the respective coordination number distributions (CNDs), local density corrected three-body correlations g$^3$ [112] as well as the oxygen–ion–oxygen angular distributions functions (ADFs) have been analysed.

Typically, the coordination number distribution and the associated average coordination number (CN) of a given solvation shell are determined based on a radial cutoff criterion coinciding with minima determined from the ion–oxygen radial distribution function $g(r)$. However, such a simple $g(r)$-based cutoff (GC) criterion is not always an adequate choice. First, the determination of a shell boundary can be to some extent ambiguous, since it may be influenced by noise and depends inter alia on the type of normalisation applied in the RDF analysis [113]. Moreover, a strictly radial criterion may exclude ligands, that should be considered as part of the first coordination sphere, while on the other hand molecules inside the cutoff should not be included as they have to be considered "blocked" by other, closer ligands.

The relative angular distance (RAD) algorithm [113,114] provides a simple and effective framework to classify ligands in the first solvation shell that is not dependent on any predefined cutoff radius. Instead, the list of potential first shell ligands is evaluated based on their increasing distance from the solute $i$. A molecule $j$ is considered as part of the first solvation layer if it remains unblocked by all previously identified closer ligands $k$ according to the criterion

$$\frac{1}{r_{ij}^2} > \frac{1}{r_{ik}^2} \cos\left(\theta_{jik}\right) \tag{1}$$

with $r_{ij}$ and $r_{ik}$ being the respective ion–ligand pair distances and $\theta_{jik}$ is the associated solvent–solute–solvent angle. As soon as the first ligand $j'$ violates the RAD criterion, it is considered as blocked and the search is ended. Thus, all unblocked ligands closer than

molecule $j'$ are considered as part of the first solvation shell in this respective configuration. Averaging over all configurations of the simulation trajectory then yields $CN_{RAD}$. In the original work, the molecules' centres of mass have been considered [113]. However, in this article the analysis was based on the oxygen atoms in order to make the results for $CN_{RAD}$ directly comparable with those of the radial cutoff $CN_{GC}$. Since the oxygen atom of a water molecule is always in very close vicinity to the respective center of mass, this approximation is not expected to have any profound impact on the determination of the coordination numbers and their distributions.

The structural properties of the first shell water molecules are of particular interest as well. To analyse the deviation from the ideal geometry, the two hydrogen atoms of a water molecule have been classified based on their orientation in the solvation complex. The H-atom directly coordinated to the solute via hydrogen bonding is considered as proximal ($H_1$) while the one located at larger distance is referred to as distal ($H_2$). Based on this classification, the associated O–$H_1$ and O–$H_2$ radial distributions as well as the corresponding ion$\cdots H_1$–O and ion$\cdots H_2$–O angular distributions have been analysed.

The mean ligand residence time $\tau$ (MRT) within the first shell has been characterised based on a direct measure of the exchange events [111]. In this approach, the MRT-value of a specific solvation shell is determined as

$$\tau_{0.5} = \frac{t_{sim} \cdot CN_{av}}{N_{0.5}} \tag{2}$$

where $t_{sim}$ is the total simulation time, $CN_{av}$ is the average coordination number of the respective solvation layer and $N_{0.5}$ corresponds to the total number of border crossing event to/from this solvation shell lasting for a minimum excursion time $t^* \geq 0.5$ ps [111]. The associated rate coefficient $R_{ex}$ is given as the ratio between the number of all border crossing attempts $N_{0.0}$ for $t^* = 0$ and $N_{0.5}$.

## 3. Results

In the following, the results obtained for the target systems via classical and QM/MM MD simulations are discussed. The key quantities are listed in Table 2, including a comparison to other theoretical and experimental investigations. Overall, a broad variation in the determined quantities can be identified, clearly highlighting the challenging nature associated to the characterisation of anionic solvation. The experimental studies by Wallen et al. [83] and Soper et al. [81] also point towards a quite sensitive concentration dependence, in particular with respect to the observed coordination number. This, on the other hand, might prove to be a limiting factor when comparing to theoretical results based on the Car-Parrinello MD framework, which especially in the past were typically limited to small system sizes, often encompassing in addition to the solute only about 30 to 128 water molecules. This and the oftentimes limited accuracy of DFT functionals at the GGA (generalised gradient approximation) level make it difficult to judge the quality of the results obtained via CPMD simulations. The latter is particularly true whenever hydrogen bonded systems are to be investigated. Simulation data on pure water have shown that the results can be notably improved if a corrective treatment of dispersive contributions is considered [44,45,50]. The latter is confirmed in a study by Bankura and coworkers comparing CPMD simulation data [78] with and without the application of the Tkatchenko–Scheffler van-der-Waals (TS-vdW) correction [115]. The same study also highlighted that the simulation results obtained using the Perdew–Burke–Enzerhofer (PBE) GGA-type functional can be notably improved if the more accurate yet more demanding hybrid variant PBE0 is applied.

**Table 2.** Maximum, avarage and minimum distances $r_\mathrm{M}^1$, $\langle r_\mathrm{M}^1 \rangle$ and $r_\mathrm{m}^1$ of the first solvation shell in the ion–O RDF in Å, average first shell coordination number determined via a g(r)-based cutoff $\mathrm{CN}_\mathrm{GC}^1$ as well as the relative angular distance approach $\mathrm{CN}_\mathrm{RAD}^1$, first shell mean ligand residence time $\tau_1$ in ps, number of registered ligand exchange events $N_{0.5}^1$ ($t^* \geq 0.5$ ps) and associated rate coefficient $\mathrm{R}_\mathrm{ex}^1$ obtained for aqueous $F^-$, $Cl^-$ and $Br^-$ via classical (MM) and RIMP2-based QM/MM MD simulations in comparison to data reported in the literature.

| | | $r_\mathrm{M}^1$ | $\langle r_\mathrm{M}^1 \rangle$ | $r_\mathrm{m}^1$ | $\mathrm{CN}_\mathrm{GC}^1$ | $\mathrm{CN}_\mathrm{RAD}^1$ | $\tau_1$ | $N_{0.5}^1$ | $\mathrm{R}_\mathrm{ex}^1$ | |
|---|---|---|---|---|---|---|---|---|---|---|
| $F^-$ | MM MD | 2.59 | 2.63 | 3.25 | 6.2 | 6.3 | 15.4 | 10 | 5.4 | this work |
| | RIMP2/MM MD | 2.46 | 2.68 | 3.36 | 4.9 | 5.2 | 1.1 | 115 | 5.2 | this work |
| | MM MD | 2.53 | | | | 5.8 ± 0.1 | | | | Ref. [73] |
| | HF/MM MD | 2.68 | | | | 4.6 ± 0.2 | | | | Ref. [73] |
| | BLYP CPMD | 2.66 | | | | 5.1 | | | | Ref. [75] |
| | BLYP CPMD | | 2.7 | 3.4 | | | | | | Ref. [77] |
| | NDIS KF/D$_2$0 1.2:100 | 2.54 | | 3.27 | | 6.9 | | | | Ref. [81] |
| $Cl^-$ | MM MD | 3.26 | 3.35 | 3.93 | 7.6 | 7.8 | 4.2 | 46 | 6.7 | this work |
| | RIMP2/MM MD | 3.23 | 3.48 | 4.16 | 8.1 | 7.5 | 1.6 | 178 | 4.5 | this work |
| | MM MD | 3.15 | | | | 5.9 ± 0.1 | | | | Ref. [73] |
| | HF/MM MD | 3.24 | | | | 5.9 ± 0.1 | | | | Ref. [73] |
| | HF/MM MD | 3.25 | | 3.9 | | 6.8 | 2.0 | | 4.6 | Ref. [74] |
| | PBE-D3 CPMD | 3.14 | | 3.78 | | 6.0 | | | | Ref. [79] |
| | PBE0-D3 CPMD | 3.17 | | 3.85 | | 6.1 | | | | Ref. [79] |
| | SCAN CPMD | 3.17 | | 3.85 | | 6.7 | | | | Ref. [79] |
| | PBE CPMD | 3.11 | | 3.64 | | 5.5 ± 0.2 | | | | Ref. [78] |
| | PBE+TS-vdW CPMD | 3.14 | | 3.78 | | 6.3 ± 0.9 | | | | Ref. [78] |
| | PBE0 CPMD | 3.14 | | 3.72 | | 5.8 ± 0.7 | | | | Ref. [78] |
| | PBE0+TS-vdW CPMD | 3.16 | | 3.73 | | 6.3 ± 0.8 | | | | Ref. [78] |
| | EXAFS NaCl 40 mM | 2.91/3.11 | | | | 4+3 | | | | Ref. [82] |
| | NDIS KCl/D$_2$O 1.2:100 | 3.14 | | 3.78 | | 7.1 | | | | Ref. [81] |
| $Br^-$ | MM MD | 3.33 | 3.45 | 4.05 | 8.1 | 8.1 | 3.0 | 313 | 4.7 | this work |
| | RIMP2/MM MD | 3.31 | 3.68 | 4.30 | 9.1 | 7.4 | 0.9 | 390 | 2.9 | this work |
| | MM MD | 3.27 | | 3.9 | | 7.6 ± 0.5 | 2.6 | | | Ref. [76] |
| | BLYP CPMD | 3.33 | | 3.9 | | 6.5 ± 0.3 | 5.7 | | | Ref. [76] |
| | XAFS/MM MC YBr$_3$ 0.3M | | 3.44 ± 0.07 | | | 6 ± 0.5 | | | | Ref. [80] |
| | XAFS RbBr 0.2M | 3.35 | | | | 7.1 ± 1.5 | | | | Ref. [83] |
| | XAFS RbBr 1.5M | 3.36 | | | | 7.2 ± 0.4 | | | | Ref. [83] |
| | XAFS RbCl 0.5 mM | 3.26 | | | | 10 | | | | Ref. [82] |
| | NDIS KBr/D$_2$O 1.2/100 | 3.32 | | 3.90 | | 6.7 | | | | Ref. [81] |

The above-mentioned shortcomings (i.e., the small number of solvent molecules and a quantum chemical treatment typically limited to GGA-DFT level) can be overcome using QM/MM MD type protocols. However, in the existing literature, only simulation studies employing the Hartree–Fock (HF) level of theory have been presented for aqueous $F^-$ and $Cl^-$ [73]. While being in a strict sense an ab initio method, HF theory suffers from the inability to take electron correlation into account, which especially in the case of hydrogen bonded systems might prove as a limiting factor. The QM/MM MD simulation protocol applied in this study aims at overcoming the above-mentioned limitations by treating the target systems at a correlated ab initio level while at the same time providing a sufficient number of solvent molecules represented via the newly parameterised SPC-mTR2 water model [70].

A number of key features of the hydration complexes can be directly extracted from the respective ion–solvent radial distributions functions depicted in Figure 2. In addition to the quantification of the average coordination number and ion–ligand distances, the differences between the purely classical and QM/MM-based simulations proved to be of particular interest. Overall, the structural properties of the solvation complexes are as expected with the main peak observed in the ion–O interaction being located in between two separate

peaks in the corresponding ion–H RDF. Based on the separation of the first peaks, the hydrogen bonded nature of the solute–solvent interaction can be directly identified. While, in principle, patterns with similar ion–water distances are observed when comparing the MM and QM/MM simulation results, a number of notable differences can be identified, foremost being a dramatic decrease in peak intensity when changing from the MM to the QM/MM description by a factor of approx. 3 for $F^-$ to about 2 in case of $Br^-$. This implies that the MM description is notably overstructured despite showing similar ion–H and ion–O distances. As outlined above, great attention has been given to identify suitable potential parameters resulting in an adequate description of the system. When comparing the peak maxima observed in the ion–oxygen RDFs determined via classical MD simulations employing the different Lennard–Jones potentials (see Figure 1b), it can be seen that it is not possible to identify a parameter set that results in an adequate ion–O distance while at the same time resulting in a reduced peak intensity. This demonstrates that the dramatic overstructuring is an inherent shortcoming of the simplified, pairwise-additive description. This trend has already been observed in previous simulation studies of the monovalent cations $Li^+$, $Na^+$ and $K^+$, albeit to a much lesser extent. Not surprisingly, subtle effects associated to the (re)distribution of the electron density such as polarisation, charge-transfer and many-body contributions are much more dominant in the anionic case.

The latter not only leads to the observed reduction in the RDF peak intensities, but also to a very pronounced tailing of the first shell ion-O peaks towards larger distances. This again demonstrates the difficulty in determining a suitable boundary for the QM treatment, which commonly is derived based on preceding classical simulations. Also in this case, simulations of cationic solutes show a much better agreement between the MM and QM/MM case.

Simulation data depicted in the form of RDF plots provide direct access to the most populated distances coinciding with the maxima of the respective RDF peaks. In contrast, experimentally determined ion–oxygen distances can be assumed to measure the associated average over the entire solvation shells of a large number of complexes present in solution as for instance explicitly highlighted by Merkling and coworkers [80]. While the difference between the latter is oftentimes negligible when characterising cationic hydrates, the strong tailing of the first shell peak may result in a notable increase of the average distance. In order to characterise the latter, the respective maximum distances $r_M^1$ are compared to their respective average $\langle r^1 \rangle$ determined as weighted average up to the border of the first solvation shell given as $r_m^1$:

$$\langle r^1 \rangle = \frac{\sum\limits_{r=0}^{r_m^1} r \cdot g(r)}{\sum\limits_{r=0}^{r_m^1} g(r)} \tag{3}$$

The resulting values obtained for the MM and QM/MM MD simulations are listed in Table 2. While in case of the classical simulations, $\langle r^1 \rangle$ is only marginally increased compared to the respective maximum distance $r_M^1$, a notable shift to larger values can be observed in the QM/MM simulations. In the latter case, the respective differences between the maximum and average first shell distance amount to 0.22, 0.25 and 0.37 Å for $F^-$, $Cl^-$ and $Br^-$, respectively, while in case of the classical simulations, much smaller deviations of 0.04, 0.09 and 0.12 Å have been determined.

The different interaction characteristics also have a notable impact on the first shell coordination numbers determined using the first shell boundary $r_m^1$ as cutoff criterion yielding $CN_{GC}$. While in the case of fluoride a decrease in $CN_{GC}$ from 6.2 to 4.9 has been observed when changing from the MM to the QM/MM description, in both the chloride and bromide system an increase from 7.6 to 8.1 and from 8.1 to 9.1 was observed, respectively. The non-integer coordination numbers discussed above result from the averaging over different microstates observed along the simulations. Comparison of the calculated coordination numbers to other experimental and theoretical data reveals that

the $CN_{GC}$ values show notable deviations especially in case of $Cl^-$ and $Br^-$. While the extraction of partial coordination patterns from measured data may prove as particularly challenging, the deviations in the coordination numbers might as well be the results of an ill-defined assignment of first shell ligands based on a radial cutoff criterion. The latter can be expected to also be negatively influenced by the pronounced tailing observed in the ion–water pair distributions resulting from the QM/MM MD simulations.

The relative angular distance (RAD) procedure by Higham et al. provides an alternative characterisation of first shell ligand distributions, which is solely based on the comparison between individual ion–ligand distance contributions weighted by the cosine of the associated solute–solvent–solute angle. The associated coordination numbers $CN_{RAD}$ determined without any radial cutoff criterion are compared to the corresponding $CN_{GC}$ values in Table 2. Although in the case of the MM simulations the RAD procedure resulted in highly similar average CN values, the results of the QM/MM simulations are greatly improved after application of the RAD criterion. While in case of $F^-$ an increase from 4.9 to 5.2 has been observed, the average coordination number decreased from 8.1 to 7.5 and from 9.1 to 7.4 in case of $Cl^-$ and $Br^-$, respectively. Thus, in all cases, the $CN_{RAD}$ values are in much better agreement with the literature compared to the $CN_{GC}$ results.

To further analyse the composition of the first solvation shell, the corresponding coordination number distributions (CNDs) as obtained from the GC and RAD analysis have been evaluated (see Figure 2). As in the case of the RDF plots, the CNDs show a notable decrease associated to a broadening of the distribution when changing from the MM to the QM/MM description in all cases. Comparison of the QM/MM-based CND plots obtained using the GC- and RAD-based ligand assignment again reveals a trend to lower coordination numbers in the latter case as already observed for the average coordination numbers. In contrast, highly similar CND plots are obtained in the MM case when comparing the two different analysis protocols.

The in-depth characterisation of the first shell coordination numbers and their respective distribution over the course of the individual simulations provides a number of important conclusions: First, the data highlight that a simple radial criterion is not always an adequate choice to determine the coordination number, and improved results can be obtained by the application of more elaborate frameworks such as the RAD criterion. In addition, it was found that the application of the latter only improves the results of the QM/MM MD simulations, while in case of the purely classical simulations, both the average CNs as well as the corresponding CNDs remained highly consistent. Considering the strongly tailed first shell peak of the radial distribution functions in the QM/MM case, it appears that interactions incorporating many-body contributions are prone to errors in a purely cutoff-based coordination number determination.

Comparing the more reliable RAD-based coordination numbers, it can be seen that the MM simulations yield notably higher average CN values than their QM/MM counterpart by approx. 21, 5 and 9% in case of $F^-$, $Cl^-$ and $Br^-$, respectively. Thus, specifically in the case of fluoride, a notable shift to higher coordination numbers is observed in the MM case. On the other hand, the CNDs determined from the MM and QM/MM simulations of $Cl^-$ display the best agreement, albeit also in this case a reduction of the most populated contributions (coordination numbers 7 and 8) is observed. One possible explanation for these findings could be that the applied potential parameters are ideal in the $Cl^-$ case, while for the other two systems a refinement of the interaction potentials might lead to improved results.

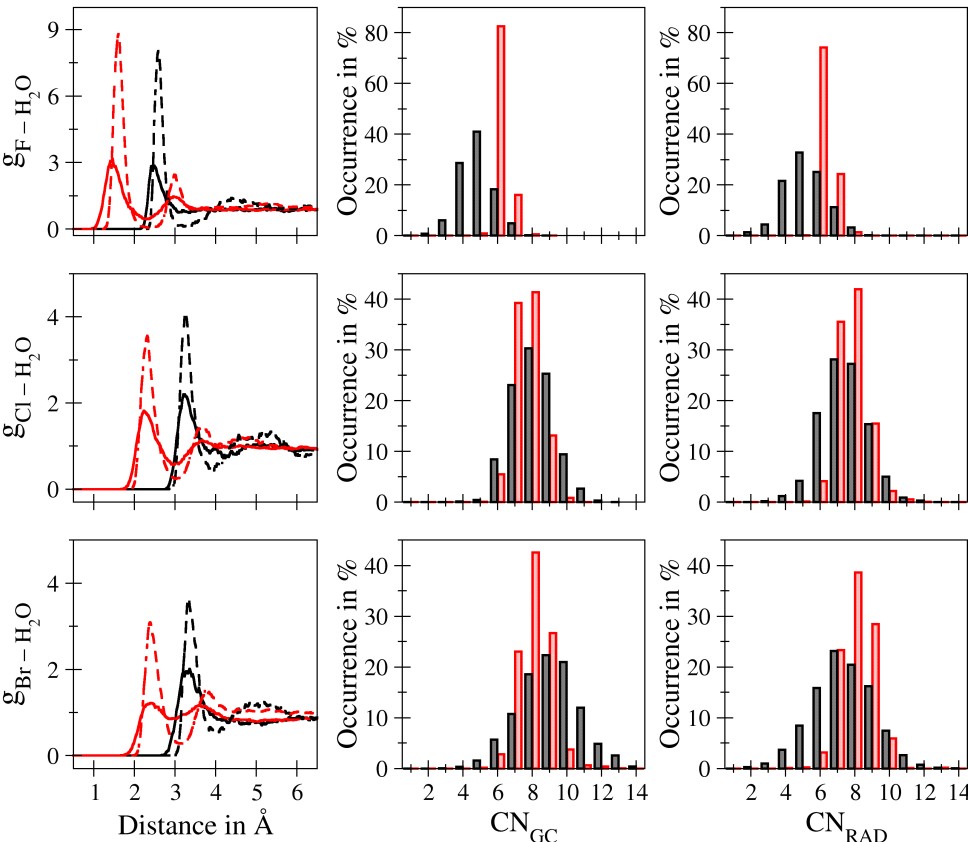

**Figure 2.** Ion–oxygen (red) and ion–hydrogen (black) radial distribution functions (left column) for aqueous F$^-$ (top row), Cl$^-$ (center row) and Br$^-$ (bottom row) determined from the QM/MM (solid line) and classical (dashed line) MD simulations along with the associated coordination number distributions based on a radial $g(r)$-cutoff criterion (center column) and the RAD analysis (right column) as obtained from the QM/MM (black) and classical (red) simulation trajectory.

Although the latter strategy can be expected to adjust the solvation parameters within a small margin, already great care has been devoted towards the identification of suitable solute–solvent Lennard–Jones potentials as outlined above. However, the pairwise nature of the classical description provides a more reasonable explanation for the observed deviations. Due to the small size of the F$^-$ ion, the hydration complex shows the smallest ion–water distances. This implies that contributions arising from the respective Coulomb interactions are highest in this case. Since the classical description does not account for many-body contributions such as polarisation and charge-transfer, it can be expected that, out of the considered systems, F$^-$ shows the highest susceptibility towards overcoordination. In contrast, the Cl$^-$ and Br$^-$ systems display notably increased first shell ion–water distances, thus reducing the magnitude of the Coulombic contributions and, thereby, the risk for observing much higher coordination numbers. Nevertheless, the high intensities in the coordination number distribution and the ion–water RDFs still points towards an overall too strong solute–solvent interaction that cannot be easily compensated by a variation in the Lennard–Jones parameters (see data in Figure 1b). While similar overcoodination effects were also observed in the case of the previously mentioned cations Li$^+$, Na$^+$ and K$^+$, the increase in the first shell ion–oxygen peak maxima amounts to at most 50%, with an overall very good agreement in the coordination number distributions when comparing the classical and QM/MM MD simulation results. These difficulties clearly demonstrate that simulations of anionic solutes are much more intricate compared to their cationic counterparts. A reduction of the net-charge of the ions in the simulations to non-integer values could offer a potential solution, which has already been discussed by different authors in the past [116–119].

In order to obtain further insight into the composition of the first solvation shell, both local density corrected three-body correlations $g^3_{I-O-O}$ measuring the O–O distance contributions within the first shell as well as oxygen–ion–oxygen angular distribution functions can be employed. While at first sight these analysis schemes appear redundant, the comparison of the different systems provided in Figure 3 demonstrates that the two different methods complement each other. For instance, based on the $g^3$-plots determined for F⁻, a similar O–O distance distribution within the first shell is observed in the MM and QM/MM simulations. However, when comparing the respective angle distributions, it can be seen that, although the O–O distances are nearly identical, the O–F–O angle contributions show remarkable differences, pointing towards a highly increased intra-shell mobility in the QM/MM case. In contrast, similar angular distributions are observed in the case of Cl⁻, while notable differences in the three-body correlations are visible. When keeping in mind that Cl⁻ displayed the best agreement in the coordination number distributions between the MM and QM/MM simulations, the capabilities of the combined $g^3$ and ADF contributions to provide further details about the solvation structure is clearly demonstrated.

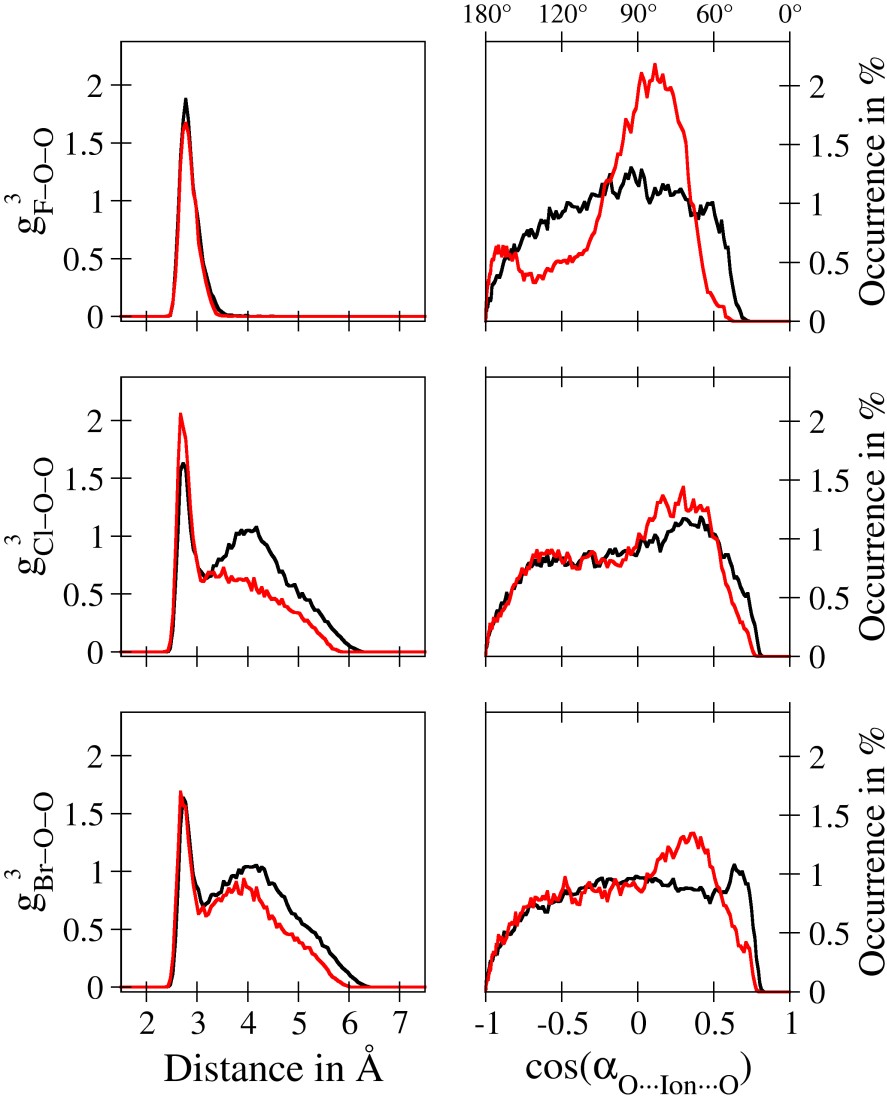

**Figure 3.** First shell ion–oxygen–oxygen three-body correlation function $g^3$ (left) and the associated O–ion–O cosine distribution functions (right) obtained for aqueous F⁻ (top), Cl⁻ (center) and Br⁻ (bottom) from the QM/MM (black) and classical (red) MD simulations. The absolute values for the O–ion–O angles follow a non-linear trend and are shown on the secondary *x*-axis.

In addition to highlighting the shortcomings of the classical description, this combined analysis provides manifold information about the differences between the target systems. For instance, in the case of $F^-$ no second neighbour distances are registered in the three-body distributions, which can be attributed to the overall low coordination number permitting only one type of neighbouring molecules. In contrast, a larger number of solvent molecules is present in the first shell of $Cl^-$ and $Br^-$, giving rise to a notable second neighbour peak located at approx. 4.2 Å. While in the case of the latter two ions, the features of the $g^3$-distributions appear similar, the respective oxygen–ion–oxygen distributions show a notable broadening especially in the $Br^-$ case, implying that in this system the highest intra-shell mobility of ligands is observed as expected.

In order to further study the ion–ligand binding properties within the first solvation shell, the associated mean ligand residence times $\tau$ (MRT) have been determined using the direct method. In this approach, all crossing events of the first shell border $r_m^1$ lasting for a minimum time of $t^*$ are counted, and the respective MRT value is then determined based on Equation (1). In addition, the respective rate coefficient $R_{ex}$ corresponding to the average number of border crossing events required to achieve one lasting ligand exchange were determined. The respective values obtained from the MM and QM/MM MD simulation trajectories of the target systems are listed in Table 2.

When considering the MRT values calculated from the classical simulations being 15.4, 4.2 and 3.0 ps in the case of $F^-$, $Cl^-$ and $Br^-$, respectively, the expected decrease in the solute–solvent interaction is well reflected. This decrease in the mean residence times is well in agreement with the associated ion–O RDF data, specifically the intensity of the minimum separating the first and second solvation shells which gradually increases from the lightest to the heaviest solute. In the case of the QM/MM MD simulations, however, a different picture emerges. First, it can be seen that the MRT values are consistently lower compared to the MM-based estimation. This is not uncommon considering the overstructured hydration discussed above, which is the result of the unpolarised, pairwise nature of the interaction in the classical model. In addition, the MRT values determined in case of $F^-$ are now slightly below that obtained for the $Cl^-$ system. Although at first sight this finding implies that the latter now displays weaker interactions with the solvent, this counterintuitive result is due to the occurrence of the average coordination number in the determination of the MRT value (see also Equation (1)). In contrast, when comparing the actual number of registered ligand exchange events lasting for a minimum excursion time of $t^* \geq 0.5$ ps, only 115 exchanges were registered in the case of $F^-$ over the sampling time of 25 ps, while 178 and 390 exchanges were registered for $Cl^-$ and $Br^-$ over the same time period, respectively. This clearly displays that the solute–solvent interaction follows the expected trend with $F^-$ forming the most stable hydrate, while the $Br^-$–$H_2O$ interaction is weakest. It should also be noted that MRT values below 1.0 ps have been associated to "structure-breaking" properties in the past [42,120]. The latter is underlined by the associated rate coefficients $R_{ex}$ measuring the average number of border crossing events to achieve one sustainable exchange event lasting for longer than $t^*$. While in case of $F^-$ on average 5.2 exchange attempts are required, notably smaller $R_{ex}$-values of 4.5 and 2.9 have been determined for the weaker hydrates of $Cl^-$ and $Br^-$, respectively. Again, small values in $R_{ex}$ have been associated to "structure breaking" solute–solvent interactions [42,120].

Although it is commonly observed that QM/MM MD simulations display an accelerated first shell ligand exchange compared to MM-based simulations, the difficulties in identifying suitable QM radii discussed above might lead to an additional acceleration of the observed exchange rates. In order to minimise QM/MM transition artifacts, it is a typical strategy to position the QM/MM boundary in a region of low density corresponding to minima in the respective ion–O distributions. As discussed above, the unexpected strong tailing observed in the ion–O RDFs of the QM/MM simulations implies that the MM-derived settings for the QM radii might not be ideal.

While in case of $Cl^-$ and $Br^-$ the ion–O RDFs still show well-defined minima in the QM/MM case, the respective first shell peak in the $F^-$ case displays only a very shallow

minimum that directly connects to the bulk without any discernable contribution indicating the presence of a second solvation shell. On the other hand, the good agreement of the first shell distance and coordination number $CN_{RAD}$ with data in the literature implies that the first shell properties are adequately represented. In order to verify that the selected size of the QM region is indeed adequate, it might prove advantageous to execute further QM/MM MD simulations with an increased size of the QM zone to monitor changes in the hydration shells and the associated mean ligand residence times. However, when considering the rather costly execution times of several months of a 10 + 25 ps QM/MM MD simulation at RIMP2 level, such a systematic adjustment of the simulation settings is quite demanding. The costly execution time of a correlated ab initio description in conjunction with the requirement to execute the MD simulations with full consideration of molecular flexibility is a further indication why QM/MM MD simulations of ionic solutes tend to be much more intricate compared to simple (i.e., monoatomic and monovalent) cations in aqueous solution.

In order to underline the requirement of taking molecular flexibility into account, the O–H bond distances observed within the first hydration shell have been analysed, thereby separating the contributions from the proximal $H_1$ atoms directly interacting with the solute from those of the distal $H_2$ atoms pointing away from the ion (see sketch in Figure 4). The comparison of the associated RDFs depicted in Figure 4 highlights the different characteristics of the O–$H_1$ and O–$H_2$ bonds. In addition to a clear separation of the associated equilibrium distances, the pronounced tailing of the O–$H_1$ bond to larger distances is visible. This is a direct consequence of the hydrogen bonded nature of the solute–solvent interaction, resulting in elongated bond distances due to the coordination to the negatively charged solute. Simulation protocols employing a rigid-body description of the solvent cannot account for these critical changes in the equilibrium geometry of the first shell ligands. On the contrary, for practical reasons, the same structural constraints have to be applied to molecules in the first solvation shell, whenever such a water model is employed to describe the solvent in the simulation system. While such a strategy might still be sufficient to derive structural and dynamical properties, simulations aiming at the determination of thermodynamic properties will be negatively influenced by the application of artificial constraints employing a reference geometry that is in a strict sense only valid for molecules in the bulk.

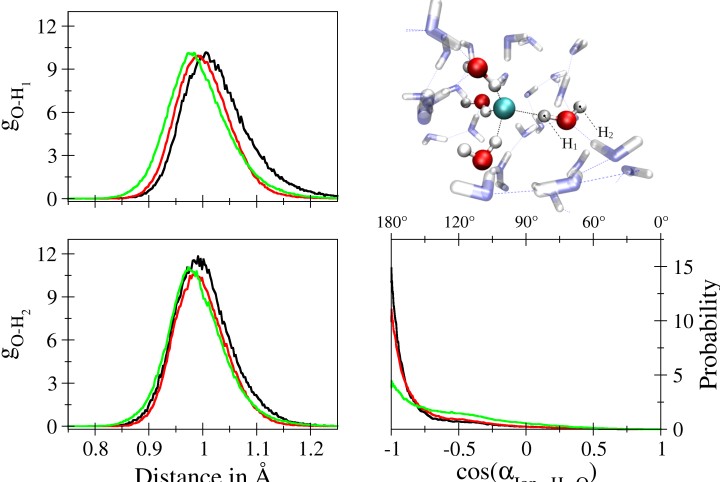

**Figure 4.** Oxygen-hydrogen radial distribution function of water molecules in the first solvation shell of aqueous $F^-$ (black), $Cl^-$ (red) and $Br^-$ (green) separated into contributions of the H-atoms in proximal ($H_1$, top left) and distal ($H_2$, bottom left) position with respect to the solute along with the associated ion$\cdots H_1$–O cosine distribution function (bottom right). The screenshot of aqueous $F^-$ displays the assignment of $H_1$ and $H_2$ for an exemplary first shell water molecule, that is re-evaluated in every simulation step.

A further interesting detail linked to the separate consideration of first shell hydrogen atoms is the distribution of the ion$\cdots$H$_1$–O angle, which is also depicted in Figure 4. While in all cases the highest contributions are observed for angles $\geq 170°$, a pronounced tailing towards lower angles is observed upon increasing size of the solute again pointing towards an increase in intra-shell mobility. In addition, a plateau near 120° is visible for all three solutes that is highest in the Br$^-$ case. This finding could either be (i) a direct consequence of intra-shell hydrogen bonding potentially promoted by the increasing ionic radii and the associated higher coordination numbers, (ii) due to contributions of ligands leaving the second solvation shell or (iii) the result of intra-shell mobility in which the ligands display a strictly dipolar, bifurcated coordination. In order to investigate the origin of this contribution in detail, a two-dimensional histogram representation correlating the ion$\cdots$H$_1$–O and ion$\cdots$H$_2$–O angles registered for each first shell molecule in the QM/MM MD simulation of aqueous Br$^-$ has been prepared (see Figure 5). It can be seen that the main contribution is again observed in the range close to 170–180°, which represents the main hydrogen bond angle already visible in the one-dimensional Br$^-\cdots$H$_1$–O angle distribution (see sketch in Figure 5, top right corner). For lower Br$^-\cdots$H$_1$–O angles a sideward V-shape is visible, with the main contributions indeed resulting from a bifurcated coordination with the Br$^-\cdots$H$_2$–O angle averaging at approx. 100° (sketch in Figure 5, top left corner). In addition, a secondary distribution is visible in which the Br$^-\cdots$H$_1$–O hydrogen bond appears to be broken while the O–H$_2$ bond is pointing away from the ion. The latter explains the very low angle of approx. 30° (sketch in Figure 5, bottom left). Thus, an overlap of two different configurations contribute to the plateau at 120° visible in the one-dimensional Br$^-\cdots$H$_1$–O distribution (Figure 4), namely, a strictly polar, bifurcated ion–water coordination plus a less populated contribution arising from configurations with broken ion–solvent hydrogen bonding most likely associated to ligand exchange. Similar distributions can be observed in the case of F$^-$ and Cl$^-$, although with significantly lower magnitude (data not shown).

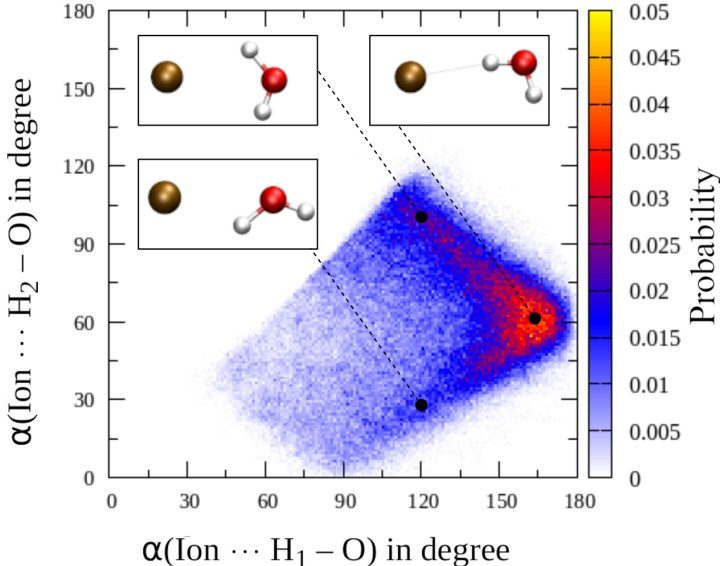

**Figure 5.** Two-dimensional histogram correlating the ion$\cdots$H$_1$–O and ion$\cdots$H$_2$–O angles registered for each first shell ligand of Br$^-$ obtained from the QM/MM MD simulation.

## 4. Conclusions

The RIMP2-based QM/MM MD simulations presented in this study provide manifold insight into the structural and dynamical properties of the target systems and provide a proof of concept for similar studies combining a correlated ab initio treatment in conjunction with full flexibility in the description of solute–solvent hydrogen bonding. The comparison to a number of previous experimental and theoretical investigations clearly highlights

the challenging nature when studying anionic solvation, being dependent on the actual concentration of studied solutions while at the same time showing a notable sensitivity on the applied level of theory. Out of all the presented calculation methods, resolution-of-identity Møller–Plesset perturbation theory of second order proved to be one of the most demanding yet most accurate calculation methods applied to this research question.

However, even in this case a number of potential shortcomings and pitfalls could be identified. In contrast to the majority of investigations available in the literature the improved description of the solute enabled by the RIMP2 framework applied to achieve comparably long sampling times of 25 ps per solute resulted in a rather unexpected tailing of the first shell peak in the ion–oxygen radial distribution functions. Typically, the boundary of the QM-treatment is selected based on similar, purely classical simulations employing suitably adjusted parameters in the applied interaction potentials. While great care was given to providing newly derived, accurate interaction parameters for the solute–solvent QM/MM coupling, the tendency towards elongated ion–solvent distances in conjunction with a notable reduction of the first shell peak intensities makes the choice of suitable QM radii to some extent ambiguous. Based on the presented simulation results, an increase in the QM/MM boundary distance determined via classical simulations appears necessary in future studies.

In addition, the strong tailing observed in the first shell ligand distributions proved problematic in the evaluation of the associated coordination numbers based on a strictly radial cutoff criterion. Application of the relative angular distance (RAD) algorithm enabling the assignment of coordinating ligands without any predefined radial cutoff distance greatly improved the results obtained for the coordination numbers. The latter only applies in case of the QM/MM MD simulations, however, while in case of the pairwise additive MM description, both the cutoff- and RAD-based ligand assignments yielded highly similar results. It can be concluded that the many-body description inherent to the quantum chemical treatment gives rise to the observed first shell tailing which in turn makes an unambiguous assignment of coordination numbers more difficult.

Despite these challenges, a consistent description of the three target systems $F^-$, $Cl^-$ and $Br^-$ could be achieved, enabling the characterisation of more complex structural properties such as three-body correlations and the distortion of first shell solvent molecules from their ideal, symmetric equilibrium geometry. In addition, the ligand exchange dynamics within the first shell could be characterised based on the simulation data, decreasing with increasing size of the solute as expected.

The presented simulation data serve as a valuable primer for advanced simulations studies, foremost in the determination of single-ion solvation free energies for which a variety of successful simulation protocols have already been implemented in case of alkaline ions. With the availability of both cationic and anionic solvation data, an inherent instrument to compare the simulation data to experimental references values determined for salt solutions becomes available, which in turn provides access to further investigations focused on absolute single-electrode potentials. Both the determination of single ion solvation properties as well as of single-electrode potentials still represents one of the few remaining questions in present day thermodynamics and the advanced QM/MM MD simulation protocol outlined in this study provides an important step towards a generalised workflow to address these prevalent research questions with the help of modern computational techniques.

**Funding:** This research received no external funding.

**Data Availability Statement:** The data that support the findings of this study are available from the corresponding author upon reasonable request.

**Acknowledgments:** The computational results presented have been achieved (in part) using the HPC infrastructure of the University of Innsbruck. Dedicated to the memory of Bernd M. Rode.

**Conflicts of Interest:** The author declares no conflict of interest.

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
