# Peer review of "Solvation Structure and Ion–Solvent Hydrogen Bonding of Hydrated Fluoride, Chloride and Bromide—A Comparative QM/MM MD Simulation Study"

_liquids, doi:10.3390/liquids2040026_

Round 1

Reviewer 1 Report

The manuscript is of good quality and of significant interest to the readers of Liquids. However, the following issues should be carefully addressed before its publication could be recommended:

Why was SPC-mTR2 used instead of flexible versions of q-TIP4P/F or TIP4P/2005f? For the non-flexible versions of water models, TIP4P performs better than SPC/E. This section of the introduction should be slightly expanded.

Although relatively short equilibration times for QMMM simulations (10ps) are an understandable consequence of the computational cost of the employed approach, it would be nice and reassuring if the author could provide some sort of measure of the relaxation time to showcase that the used equilibration time is adequate to relax the system?

In description of the QMMM approach the author stated that ion and non QM solvent interact via a potential. The potential includes the electrostatic interaction from the ionic point charge and a LJ interaction. The studied anions are known to be polarizable in certain conditions (see for example https://doi.org/10.1021/jp9512319, https://doi.org/10.1021/jp982029j ). Since this modifies the electrostatics which is known to be very long-ranged due to its slow decay could the author comment on why it is appropriate to ignore these for the ion – MM solvent part?

A possible convenient way to define the local coordination that would also be useful for this approach is the so-called radial angle distance method (DOI: 10.1063/1.4961439 , DOI: 10.3390/e21080750) which works by determining if the connection by 2 particles is blocked by another particle. Since the electrostatics are usually dominated by screening effects such an approach might be good for solving issues with ideally non-integer preferred coordination numbers mentioned in the manuscript.

Page 5 row 8: “that that” One that should be deleted

Table 1 caption: “rspective” should be “respective”

Author Response

The reply to the comments of reviewer 1 are included in the attached pdf-file, also highlighting all changes made in the manuscript.

Reviewer 2 Report

The manuscript of Hofer concerns the solvation structure around halide anions in water. It is a rather timely effort to concentrate on these species, and furthermore, I found the author's computational approach intersting. In general, I support publication, after the author has considered the issues below.

(1) The fact that the new computational results are are compared with experimental ones is very much welcome. What is less understandable is that  no comparison with X-ray diffraction data is provided (cf. Table 2), although the weighting factors of the ion-water partial contributions are much more favourable than they are for neutron diffraction. Combined neutron + X-ray diffraction studies are available for all cesium-halide solutions (by Mile et al., J. Phys. Chem. and J. Mol. Liq., around 2010). Also, a rather recent work by Pethes et al. (PCCP 2020) considers chloride ions in aqueous LiCl solutions from the point of view H-bonding: this paper seems to be also based on a combined neutron+X-ray diffraction study. (Note that I by no means require that these works must be cited, but I would be happy if the author looked at these papers and decide if they are relevant or not.)

(2) Table 2 is a very nice summary of the calculations that the reader will value. Exactly because it's so transparent, it is easy to notice that the new computations, particularly the RIMP2/MM MD ones, fail to provide a a good agreement with experiments (at least not with experimental results listed in Table 2). I aware of the difficulties with deriving partial coordination numbers from diffraction data, so I do not say that it is surely the calculations that are problematic. However, a more detailed discussion on the (dis)agreement with experimental quoted (in Table 2) would be in order.

(3) In Figures 2 and 3, angular distributions appear where the the independent variable is 'angle in degrees'. I would suggest that these curves would better appear as a functions of the cosines, which would not distort the two ends (at 0 and 180 degrees) of the distributions. Surely the angle-dependent function in Figure 2 would not not approach zero, and the H-bonding (straight) angle would be more populated in Figure 3, if cosines were used.

Author Response

The reply to the comments of reviewer 2 are included in the attached pdf-file, also highlighting all changes made in the manuscript.

Reviewer 3 Report

The manuscript “Solvation Structure and Ion-Solvent Hydrogen Bonding in Hydrated Fluoride, Chloride and Bromide – a Comparative QM/MM MD Simulation Study” by Thomas Hofer is devoted to QM/MM simulations of hydrated fluoride, chloride and bromide anions using a computationally demanding, but highly accurate RI-MP2 approach. The topic is not new and countless MM MD, QM/MM MD and ab initio MD studies of halides exist. However, the correlated treatment of electrons is not something routinely applied to aqueous ionic systems and this make the work quite interesting. Overall, I find the manuscript publishable in Liquids, provided the author responds to the comments listed below:

  1. For new developments in the field of ab initio single ion thermodynamics, the author is advised to consult the reference found in PNAS 117 (2020) 30151–30158. Also, a mention of quasi-chemical theory for ion solvation studies would be beneficial for the introduction, see for example https://arxiv.org/abs/physics/9909004.

  2. The LJ interaction parameters are derived on the basis of purely classical MM MD simulations. In the QM/MM approach, the ion will be screened by the first shell QM water molecules, so it begs the question to what extent is the MM LJ potential transferable. For example, additional tests of QM/MM cluster energetics could have been performed to additionally validate the force field. Furthermore, the selection of “best” LJ parameters is done only on the basis of strictly structural criteria (RDF/coord no.), while the reproduction of energetics might be equally important, especially in the context of single ion thermodynamics. Finally, I found Fig. 1 quite misleading and hard to follow. Maybe a better way of data presentation is possible here.

  3. The QM code applied in the simulation protocol is not mentioned and I doubt the author would choose to write his own code from scratch. I suppose the QM interactions were handled by external code and the details should be mentioned. It also concerns the MM part, unless this was done with in-house code (although a citation would be preferable even in this case, to point the reader at further details). Also, the author describes in detail the issue of QM ion–MM water interactions, but does not specify in detail, how are QM–MM water interactions handled.

  4. What were the criteria to classify a given hydrogen atom as proximal/distal to the anion? Was this simply a distance-based criterion? Figure 4 is also misleading: the curves for anions are labeled with top/bottom/etc instead of colors, the sketch is not mentioned in the caption and the angular distribution is most probably wrongly labeled (as H1–O–H2, while I suppose this should be X–H1–O).

  5. Is there any significance for the value of 0.5 ps in the definition of the minimum excursion time?

  6. I am puzzled by the sentence “While in case of Cl− and Br− the ion-O RDFs still show well-defined minima in the QM/MM case, the respective first shell peak in the F− case directly connects to the bulk with not discernible minimum.” An examination of Figure 3 reveals that at least qualitatively the minimum in the case of fluoride is as deep as for the two other halides. NB, also in the case of this figure some errors in the caption are found.

  7. The alternative explanation to the plateau around 120 deg in the X–H1–O angular distribution could be strictly dipolar coordination, i.e., H2O pointing with both hydrogens towards the anion. Is it possible to differentiate the two possibilities from the present data?

Author Response

The reply to the comments of reviewer 3 are included in the attached pdf-file, also highlighting all changes made in the manuscript.

Round 2

Reviewer 1 Report

The authors successfully resolved all issues raised by this reviewer. Consequently, the manuscript has been significantly improved and can be in its current version recommended for publication in Liquids.

Reviewer 3 Report

I am fully satisfied with the changes made to the manuscript and the very detailed response that I received from the author. I am of the opinion that the work is publishable in the present form.